# Neural substrates of treatment-resistant schizophrenia and the response to clozapine: A structural MRI study in a clinical setting

Yuto Masumo[1], Nobuhisa Kanahara[2]*, Yuji Otsuka[3], Yusuke Sudo[2], Teruyuki Ishii[4], Hirotaka Sato[4], Keita Idemoto[5], Hiroshi Komatsu[6], Yuko Fujita[2], Yasunori Oda[1], Tomihisa Niitsu[1], Yoshiyuki Hirano[7], Masaomi Iyo[1,8]

1 Department of Psychiatry, Chiba University Graduate School of Medicine, Chiba, Chiba, Japan, 2 Division of Clinical Neuroscience, Chiba University Center for Forensic Mental Health, Chiba, Chiba, Japan, 3 Department of Psychiatry, Asahi General Hospital, Asahi, Chiba, Japan, 4 Department of Radiology, Chiba University Hospital, Chiba, Chiba, Japan, 5 Department of Psychiatry, Satsuki-kai Sodegaura Satsukidai Hospital, Sodegaura, Chiba, Japan, 6 Department of Psychiatry, Tohoku University Hospital, Sendai, Miyagi, Japan, 7 Chiba University Research Center for Child Mental Development, Chiba, Chiba, Japan, 8 Department of Psychiatry, International University of Health and Welfare Narita Hospital, Narita, Chiba, Japan

* kanahara@faculty.chiba-u.jp

## Abstract

### Background

Predicting the responsiveness to clozapine among individuals with treatment-resistant schizophrenia (TRS) is difficult. A candidate predictor of clozapine response is the length of time prior to the introduction of clozapine treatment. The relationship between this measure and structural MRI findings has not been established.

### Patients and methods

We compared the cortical-volume ratio between patients with TRS (n = 40 including 20 clozapine-treated patients) and non-TRS patients with schizophrenia (n = 64) and between each of these patient groups and healthy controls (HCs). We then investigated brain regions related to both responsiveness to clozapine and the duration between the TRS designation and the introduction of clozapine.

### Results

The three-group comparison revealed that compared to the HCs, both patient groups had significantly lower cortical-volume ratios in widespread brain regions. However, there was no significant difference in the brain regions between the TRS and non-TRS groups: compared to the non-TRS group, the TRS group showed smaller volumes in a wider range of brain regions only at the uncorrected level. The correlational analysis of regions related to clozapine responsiveness did not identify any region

**Data availability statement:** All relevant data are within the manuscript and its Supporting Information files.

**Funding:** The author(s) received no specific funding for this work.

**Competing interests:** N.K. reports honoraria from Otsuka Pharmaceutical Co., Sumitomo Pharma Co., Janssen Pharmaceutical., Meiji Seika Pharma Co., Mitsubishi Tanabe Pharma Co., MSD. Y.Od. reports honoraria from Eisai Co., Otsuka Pharmaceutical Co., Meiji Seika Pharma Co. T.N. reports honoraria from Otsuka Pharmaceutical Co., Viatris Inc., Daiichi Sankyo Co., Sumitomo Pharma Co., Meiji Seika Pharma Co., and Yoshitomi Yakuhin Co. He also received research funding from NTT Precision Medicine, Inc. Y.M., Y. Ot., Y.S., T.I., H.S., K.I., H.K., Y.F., Y.H., and M.I. have no conflicts of interest to declare.

that survived the correction for multiple comparisons. No relationship between any cortical region and the length of time prior to clozapine introduction was observed.

## Conclusion

Overall, these results failed to identify the cortical region responsible for the treatment response to clozapine. The lack of correlations between the length of time prior to clozapine introduction and cortical regions might have been derived by insufficient statistical power, thus necessitating further research.

## Introduction

Clozapine (CLZ) is the only antipsychotic with established efficacy for treatment-resistant schizophrenia (TRS) [1], but CLZ has not been widely used globally to treat patients with TRS [2,3]. Patients and physicians may have some hesitation related to the several serious potential adverse events associated with CLZ treatment and the burden of the necessary regular blood monitoring [4–6]. In addition, when patients meet the diagnostic criteria for TRS but do not receive CLZ, they tend to be treated with high-dose medications (including polytherapy) [7,8]. This treatment situation increases the risks of dopamine supersensitivity psychosis (DSP) and tardive dyskinesia (TD) [9–11].

The identification of a clinical indicator that can be used to predict individual patients' responsiveness to CLZ could help achieve more favorable prognoses for patients with TRS. Although several such indicators have been examined, no definitive indicator has been identified; for example, the findings regarding the patient's age at disease onset [12] and the age at the commencement of CLZ treatment [12,13] are not consistent across studies [14].

Several research groups have suggested that the length of time prior to the introduction of a CLZ regimen for patients with schizophrenia might be a relatively promising clinical indicator of responsiveness to CLZ, with consistent findings among the studies [13,15–18]. Such an indicator predicting CLZ responsiveness is not only useful in clinical practice; it also leads to a hypothesis that the continuity of severe psychopathology (i.e., TRS) can be accompanied by an accumulation of structural and functional damages in patients' brains, contributing to lowered CLZ efficacy, even though this agent is introduced as a final resort.

This problem is very relevant in clinical practice since the optimal timing for the introduction of CLZ to achieve a better prognosis has not been established. Most patients with schizophrenia first receive CLZ several years after they meet the TRS criteria mentioned above. It is possible that a prolonged exposure to glutamine neurotoxicity or dopamine supersensitivity could negatively affect patients' brain substrates, but the mechanisms underlying this are not fully understood. Indirect findings in patients have indicated that the duration of relapse episodes or the treatment intensity after a relapse of psychosis could lead to deficits in brain volumes [19–21]. These data also suggested that a longer period of time before the introduction of

CLZ treatment could have negative impacts on patients' brains due to the development of unstable psychosis and/or the inability of high-dose non-CLZ antipsychotic medication to control psychosis. However, our search of the relevant literature identified no reports of potential links between neuronal substrates in TRS patients and the length of time prior to their commencement of treatment with CLZ.

The magnetic resonance imaging (MRI) studies of TRS patients treated with CLZ are limited to only a few reports: these studies concerned the frontal lobe [22–24], parietal lobe [25], occipital lobe [24,25], and temporal lobe [22,24–26], and the studies' findings are not in agreement. This uncertainty is attributable to the small size of the sample in each study and other confounding factors inherent to the patient characteristics such as longer disease course, complex psychopathology, high-dose medication regimens, and other variables. To date, most of the MRI studies on responsiveness to CLZ were of patients who had already been treated with CLZ, adding further confounding effects since CLZ can affect the brain structure by itself [19] We thus speculated that the prediction of CLZ responsiveness based on structural MRI findings that were obtained prior to the patients' CLZ introduction and the identification of clinical indicators such as the length of time prior to CLZ introduction would be valuable in clinical practice.

To test this speculation, we conducted the present study, via the following three steps. (1) We investigated structural differences in the brain between a group of patients with TRS and a group of patients with non-TRS conditions in our clinical setting; (2) for the patients with TRS who were subsequently treated with CLZ, we conducted both a dichotomous investigation (i.e., a between-group comparison) of brain regions that may be associated with CLZ responsiveness and a correlational analysis, and (3) we sought to identify cortical areas related to the length of time prior to CLZ introduction. Our hypothesis was that (*i*) no great difference in the cortical volume would be observed between the TRS and non-TRS patients, due to the relatively small size of our sample, but (*ii*) there would be brain regions with significant relationships with the length of time prior to CLZ introduction.

## Patients and methods

### Ethical review

This study was approved by the ethics review committees of the respective medical institutions (Chiba University, approval no. 1189; Asahi General Hospital, approval no. 2022031506). MRI images covered by the study were conducted during the period from January 1, 2005 to March 31, 2019. The data access period (including medical information that identified individual participants) was from December 1, 2019 to March 31, 2023 at Chiba University and from April 1, 2022 to December 4, 2022 at Asahi General Hospital.

Regarding the requirement for patients' informed consent, the detailed information of the study was provided on our department's website and gave the patients an opportunity to consider participation in the study on an opt-out basis. The study followed the Ethical Guidelines for Medical and Biological Research Involving Human Subjects in Japan.

### Patients

The inclusion criteria were (*i*) status as an inpatient or outpatient who had been diagnosed as having schizophrenia or schizoaffective disorder based on the Diagnostic and Statistical Manual of Mental Disorders, Fifth edition (DSM-5) and were treated at either of the study hospitals, and (*ii*) having undergone a head MRI examination during the above-described period. We divided the candidate patients into those with TRS and those with non-TRS, based on the operational diagnostic criteria for TRS: the patients who did not show sufficient improvement in psychotic symptoms despite treatment with ≥ 600 mg of the chlorpromazine equivalence (CP-equivalent) of two or more different antipsychotics for ≥ 4 weeks of each trial, and whose Global Assessment of Functioning (GAF) score did not exceed 41 points in the year prior to the introduction of CLZ (i.e., the 'no-response' criteria). For the TRS patients who had been treated with CLZ, since our study's main purpose was to examine brain regions related to responsiveness to CLZ, we included the MRI images

of patients who were scanned within 2 months before the CLZ commencement. For the other patients (i.e., the non-TRS patients and the patients with TRS but medicated with non-CLZ agents), we collected their MRI images if their medication was stable for ≥ 1 month prior to the MRI examination. **Fig 1** is the flow chart of the study's analyses.

We excluded patients: (*i*) whose reason for CLZ introduction met the 'intolerance to antipsychotics' criteria, (*ii*) who had psychiatric disorder(s) other than schizophrenia or schizoaffective disorder, (*iii*) with significant organic brain disease or epilepsy, or (*iv*) who had a serious physical disease, and (*v*) patients who were considered inappropriate for the study by their physicians.

## Assessments

**Indicators at the time of head MRI imaging.** The patients' clinical information was collected from their medical records: sex, age at the MRI examination, age at disease onset, and number of hospitalizations prior to the MRI examination. The antipsychotic medication each patient was taking at the time of the MRI examination was also extracted, and the medication doses were converted to the CP-equivalent dose [27,28].

**Indicators of the patients' responsiveness to CLZ.** The following psychopathological assessments were administered only to the patients who were being treated with CLZ: the length of time before the introduction of CLZ treatment (defined as the length of time from the timepoint of the establishment of TRS to the timepoint of CLZ induction), the Brief Psychiatric Rating Scale (BPRS) [29], the GAF scale, and the Clinical Global Impression-Severity (CGI-S) in the month prior to the CLZ introduction; in addition, the GAF, CGI-S, and Clinical Global Impression-Change (CGI-C) scale at 1 year after the introduction of CLZ. The duration of illness was calculated from the onset of illness to the CLZ induction.

Regarding the patients' responsiveness to CLZ, 'responder' was defined as a ≥20-point change in the patient's GAF score from baseline to 1 year of treatment. The patients whose GAF score changes were <20 points were classified as non-responders. These were obtained by the patients' physicians as routine in the clinical practice.

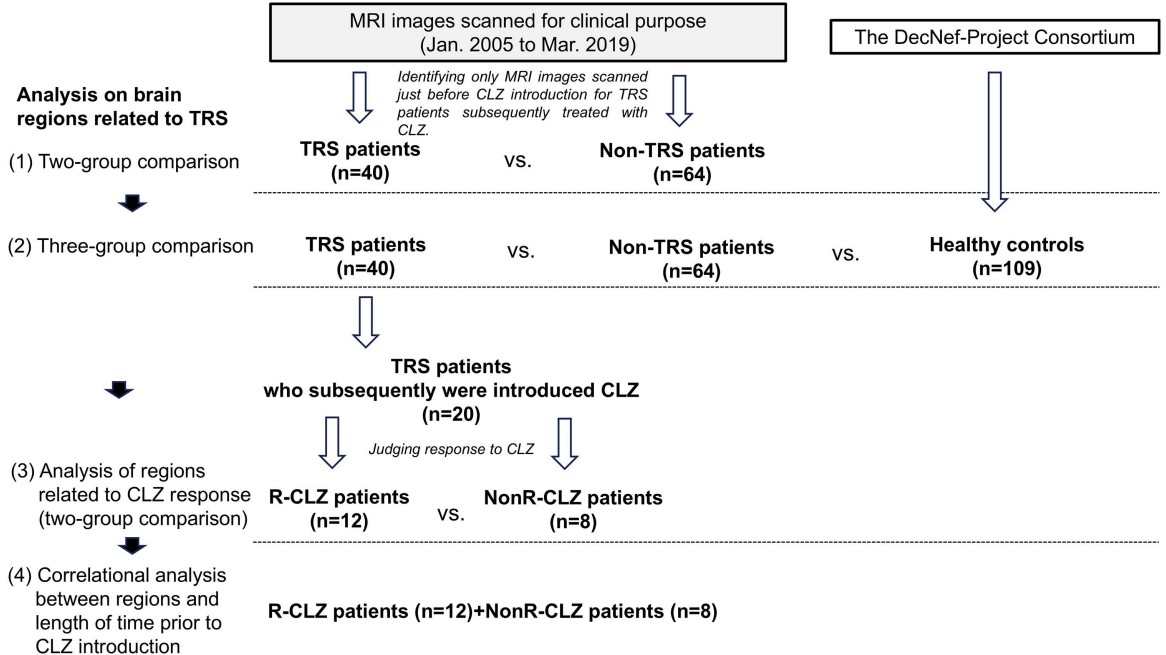

**Fig 1. Flow chart of the present study.**

## Image capture and analysis

The patients' three-dimensional T1-weighted images were obtained with the use of five different MRI scanners at Chiba University Hospital (protocol MRI-1, n = 55; MRI-2, n = 20; MRI-3, n = 21; MRI-4, n = 7) and Asahi General Hospital (MRI-5, n = 1). (S1 Table) The structural MRI data were acquired by covering the whole brain with a sagittal 3D-magnetization-prepared gradient echo sequence. The imaging parameters are in S1 Table.

For the MRI images of healthy controls (HCs), we leveraged the dataset of brain MRI images of healthy Japanese adults in the DecNef-Project Consortium (http://www.cns.atr.jp/decnefpro/). A total of 109 images (protocol MRI-6, n = 98; MRI-7, n = 11; S1 Table) were selected so that the HCs' ages and sexes were matched with those with the patients.

We excluded manually obtained images in which the entire brain was not imaged appropriately or showed clear artifacts. The image analysis was performed using the FreeSurfer 7.1.1 program (https://surfer.nmr.mgh.harvard.edu/). We also excluded images with a voxel registration error by conducting a visual quality assessment. The cortical volume per brain region was calculated based on the Desikan-Killiany (DK) atlas cited below. We did not compare the cortical thickness values, but we compared the groups' cortical gray matter-volume ratios (i.e., the ratio of the regional volume/intra cranial volume [ICV]; we used the estimated total intracranial volume [eTIV] in FreeSurfer), which are more appropriate for the comparisons of structures between the patient groups and HC group since the images were obtained with five different MRI scanners [30]. We used the DK atlas to visualize the results [31].

## Statistical analyses

For the patients' demographic and treatment-related information, Student's t-test and analysis of variance (ANOVA) were applied to continuous variables in the two-group and three-group comparisons, respectively. The $\chi^2$ test was applied for categorical variables. The threshold of statistical significance for these comparisons was set at $p < 0.05$.

For the analysis of brain cortical volumes, we performed an analysis of covariance (ANCOVA) with age, sex, and the MRI scanner as covariates for the two- and three-group comparisons since the patients' images were obtained with five different MRI scanners. A partial correlational analysis with age, sex, and the MRI scanner as covariates was performed to investigate the relationship between each clinical indicator and the cortical-volume ratio of each segment. For multiple comparisons, we applied the Benjamini-Hochberg procedure with the false discovery rate (FDR) at $p = 0.05$ for a total 68 subregions, as FreeSurfer divides the brain atlas into a total of 34 subregions within each hemisphere. All of the statistical analyses were performed using SPSS ver. 29 (IBM, New York, NY).

## Results

### Demographic information of the TRS and non-TRS groups

Magnetic resonance images were extracted from 104 patients (TRS patients, n = 40; non-TRS patients, n = 64). A total of 109 magnetic resonance images of healthy Japanese individuals were extracted from the DecNef-Project Consortium dataset. As shown in Table 1, the sex and age at the MRI examination did not differ among the TRS, non-TRS, and HC groups. The age at illness onset in the TRS group was significantly younger than that in the non-TRS group. The antipsychotic dose at the MRI examination in the TRS group was significantly higher than that in the non-TRS group (p < 0.001).

Within the TRS group, 20 patients (12 females, eight males) had been under treatment with CLZ. Although there were no significant differences in demographic measures between the patients under subsequent CLZ treatment and the patients being treated with other antipsychotics, significant differences were observed for treatment-related measures: the number of hospital admissions and the antipsychotic dose at the MRI examination in the patients treated with CLZ were significantly higher than those in the patients treated with non-CLZ antipsychotics (both p < 0.05) (S2 Table). Treatment-related information of the patients with TRS who were treated with CLZ or non-CLZ agents is presented in S2 Table and described in detail in the **Supplementary Material (S1 File)** (*1. Treatment outcomes in the patients treated with CLZ*).

**Table 1. The patients' and healthy controls' demographic and treatment information.**

| | TRS n = 40 | Non-TRS n = 64 | HC n = 109 | Statistic values |
|---|---|---|---|---|
| Males/females, n | 16/ 24 | 26/ 38 | 45/ 64 | $\chi^2 = 0.022$, p = 0.989 |
| Age at MRI examination, yrs | 39.68 (13.34) | 40.11 (14.15) | 38.18 (12.47) | $F = 0.489$, p = 0.614 |
| Diagnosis of schizophrenia/schizoaffective disorder, n | 38/ 2 | 51/ 13 | | $\chi^2 = 4.676$, p = 0.031 |
| Age at illness onset, yrs | 23.24 (8.82) | 28.64 (12.68) | | $t = -2.560$, p = 0.012 |
| Age at TRS establishment, yrs | 38.59 (12.67) | | | |
| Antipsychotic dose at MRI examination (CP-eq.), mg/day | 844.40 (616.01) | 434.45 (362.77) | | $t = 3.816$, p < 0.001 |

The data are n or the average (std. dev.).

CP-eq.: chlorpromazine-equivalent, HC: healthy control, TRS: treatment-resistant schizophrenia.

## Comparison of the cortical volume in the TRS and non-TRS patients

**Comparison of the TRS and non-TRS patients.** As shown in **S3 Table**, the comparison of cortical volume ratios between the TRS and non-TRS groups controlling for age, sex, and the MRI scanner did not detect a significant difference at the statistical threshold after FDR correction. At the uncorrected level, however, this analysis identified only one region, the left frontal pole, the volume of which was significantly larger in the TRS group compared to the non-TRS group ($F = 4.610$, p = 0.034, Cohen's $d = -0.190$).

Our comparison of cortical volumes identified 14 regions that were smaller in the TRS group relative to the non-TRS group, located mainly in the frontal and temporal lobes at an uncorrected level of significance (**Fig 2**): the bilateral lateral frontal cortices (the left caudal middle frontal, bilateral rostral middle frontal, and right superior frontal cortices), the medial frontal cortices (left caudal anterior cingulate [cACC]), the bilateral temporal cortices (the bilateral bank of the superior temporal sulcus [STS], and the right transverse temporal, bilateral superior temporal, and bilateral middle temporal cortices), the right lateral occipital, and the medial occipital cortex (i.e., cuneus) (The right cuneus had the largest Cohen's $d$ (0.508) with $F = 7.643$, p = 0.007, as shown in **S3 Table**). When the patients' age at illness onset and the antipsychotic dose were included as additional covariates, the analysis revealed almost the same results (**S4 Table**). No regions had a significantly greater cortical volume in the TRS group compared to the non-TRS group. The difference between the results of the ratio data and those of the volume data suggests the possibility that a global structural alteration in the TRS group diluted the regional differences between the two groups.

**Three-group comparisons.** Since we lacked MRI images of the HCs, we compared the TRS and non-TRS patients with the HCs from an open dataset to confirm whether the patients' results (as described in the 'Comparison of the TRS and non-TRS patients' section) were valid as general findings of schizophrenia patients.

The ANCOVA with age, sex, and the MRI scanner as covariates revealed significant differences in only five regions at the FDR-corrected threshold: the bilateral medial orbitofrontal cortex (mOFC), the left rostral anterior cingulate cortex (rACC), the left frontal pole, and the right superior frontal gyrus.

However, the ANCOVA with age and sex as covariates (without the MRI scanner) identified a significant between-group difference in the majority of the cortical regions (56 out of 68 cortical regions) with FDR-corrected p < 0.05 as the threshold for significance: 25 regions in the frontal cortex, 14 regions in the temporal cortex, 12 regions in the parietal cortex, one region in the occipital cortex, and the bilateral insula (**S5 Table**). *Post-hoc* comparisons of the regions with a significant effect of the group revealed a lower cortical volume in all regions in the TRS and non-TRS patients compared to the HCs (**Fig 3**). Most of these regions exhibited middle-to-large effect sizes (Cohen's $d$-values >0.40), and the region with the greatest effect size was the left rACC in both the TRS group ($d = 0.988$) and the non-TRS group ($d = 0.946$) (**S5 Table**).

No significant difference was observed in any region between the TRS and non-TRS groups at the statistical threshold after FDR correction (**S5 Table**). The region with the greatest effect size ($d = -0.422$) was the right paracentral cortex.

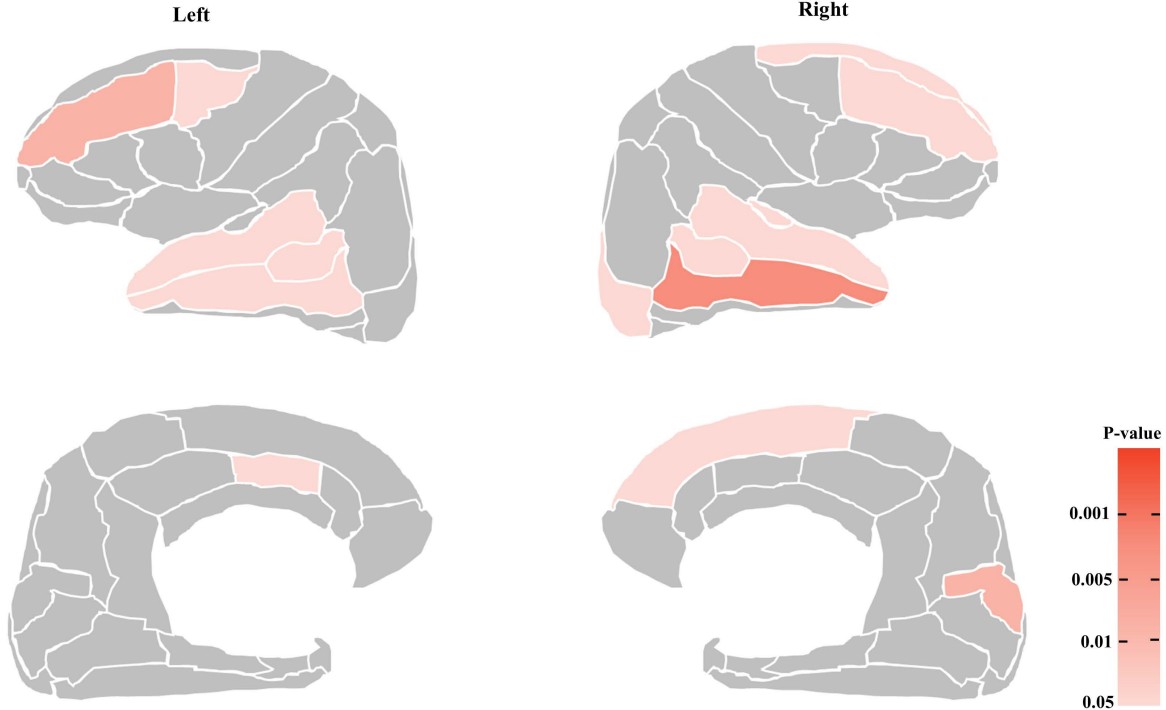

**Fig 2. The areas of greater volume reduction in the treatment-resistant schizophrenia (TRS) group compared to the non-TRS group with the significant threshold as uncorrected p < 0.05. Left panels:** the lateral (*upper*) and medial (*lower*) surfaces of the left hemisphere. **Right panels:** the lateral (*upper*) and medial (*lower*) surfaces of the right hemisphere.

Regarding the difference between the results of the analyses with and without the MRI scanner included as a covariate, we suspect that since assigned covariates for MRI scanners were separated by groups (i.e., designated 'MRI-1'-'MRI-5' to the patients and 'MRI-6' and 'MRI-7' to the HCs, respectively in **S1 Table**), any actual differences in volumes between groups likely disappear by correcting MRI scanners. In the between-group comparisons, the results of the analysis that did not include the MRI scanner as a covariate can be interpreted as the actual results. A further analysis of the potential effects of different MRI scanners on the between-group comparisons was conducted; the results are shown in the **Supplementary Material (S1 File)** (*2. Potential effects of MRI scanners within each group*).

**Brain regions related to the response to CLZ**

   **Comparison of cortical volume between the CLZ responders and non-responders.**  Since this comparison included a smaller number of patients than initially planned (R-CLZ, n = 12; NonR-CLZ, n = 8), the results are provided in the **Supplementary Material (S1 File)** (*3. Comparison of cortical volume between the CLZ responders and non-responders*) and **S6 Table**. Briefly, no significantly different cortical volume region was observed at the statistical threshold after FDR correction in the multiple comparison.

   **Correlations of the cortex volume with clinical measures relevant to CLZ responsiveness.**  Regarding the patients' responsiveness to CLZ treatment, no significantly different region was observed at the statistical threshold after FDR correction in the multiple-group comparison. As illustrated in **Fig 4** (with the r-values and p-values listed in **Table 2**), at an uncorrected level, the change in the GAF score showed a significant positive correlation with the right cACC, left pars orbitalis, bilateral pericalcarine, and the right cuneus. The CGI-C scores showed a positive relationship with the left cACC, rACC, pars orbitalis, pars triangularis, left pars opercularis, and right superior frontal. No significant correlations

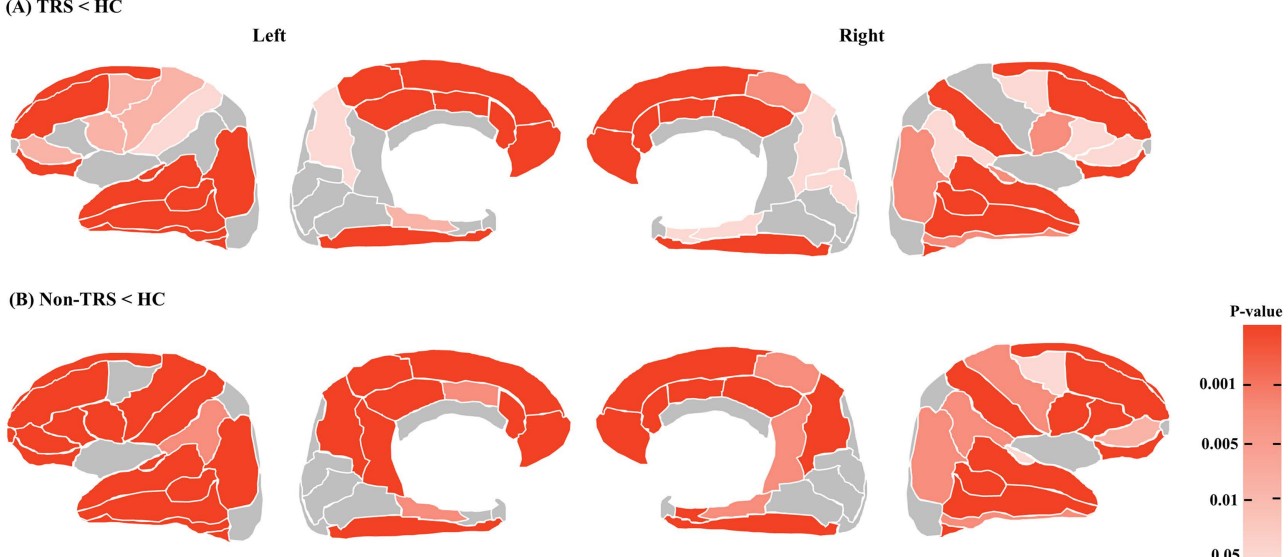

**(A) TRS < HC**

Left Right

**(B) Non-TRS < HC**

P-value

0.001
0.005
0.01
0.05

**Fig 3. The areas with a reduction of the regional volume ratio in the TRS and non-TRS groups at the whole brain level revealed by the ANCOVA for the TRS, non-TRS, and HC groups with the significance threshold as p<0.05 with FDR correction. (A)** The areas with a greater reduced volume ratio in the TRS group compared to the HCs. **(B)** The areas with a greater reduced volume ratios in the non-TRS group compared to the HCs. These figures line up from left to right with the lateral and medial surfaces of the left hemisphere and the medical and lateral surfaces of the right hemisphere of the brain.

were observed in other areas. These results were almost the same when the analysis included the antipsychotic dose as a covariate (**S7 Table**).

There was no measure prior to the commencement of CLZ treatment (i.e., age at disease onset, age at the TRS diagnosis, age at the CLZ commencement, or the length of time before the introduction of CLZ) that showed a significant relationship with any cortical volume in the CLZ-treated patients at the FDR-corrected threshold, and at the uncorrected level only the patients' age at disease onset was related to two regions (left cACC, r=−0.503, p=0.047; left paracentral, r=0.502, p=0.048).

## Discussion

The main findings of this study are as follows. Compared to the healthy controls, the TRS and non-TRS patients showed greater cortical volume reductions; these were mainly in the lateral/medial frontal and lateral temporal regions, but also in the parietal and occipital regions (**S5 Table and Fig 3**). Compared to the non-TRS patients, the TRS patients did not exhibit significantly different brain regions at the FDR correction. However, we observed smaller cortical volumes in the lateral and medial frontal temporal cortices and the occipital region in the TRS patients compared to the non-TRS patients at an uncorrected level for significance (**S3**, **S4 Tables and Fig 2**).

Regarding the responsiveness to CLZ, no regions survived the FDR correction for multiple-group comparisons: we observed that several regions including the inferior frontal gyrus (IFG), ACC, and occipital cortex were related to the CLZ response in the between-group comparison or in the correlational analysis at the uncorrected level for significance (**Table 2**, **S6**, **S7 Tables and Fig 4**, **S1 Fig**). In addition, no cortical region was related to the length of time prior to CLZ introduction even at the uncorrected level, a finding that does not support our study hypothesis. Although we identified significant structural declines in the TRS and non-TRS groups, the results of this study also indicated that the negative findings concerning the regions related to the patients' responses to CLZ and the length of time prior to CLZ introduction in the

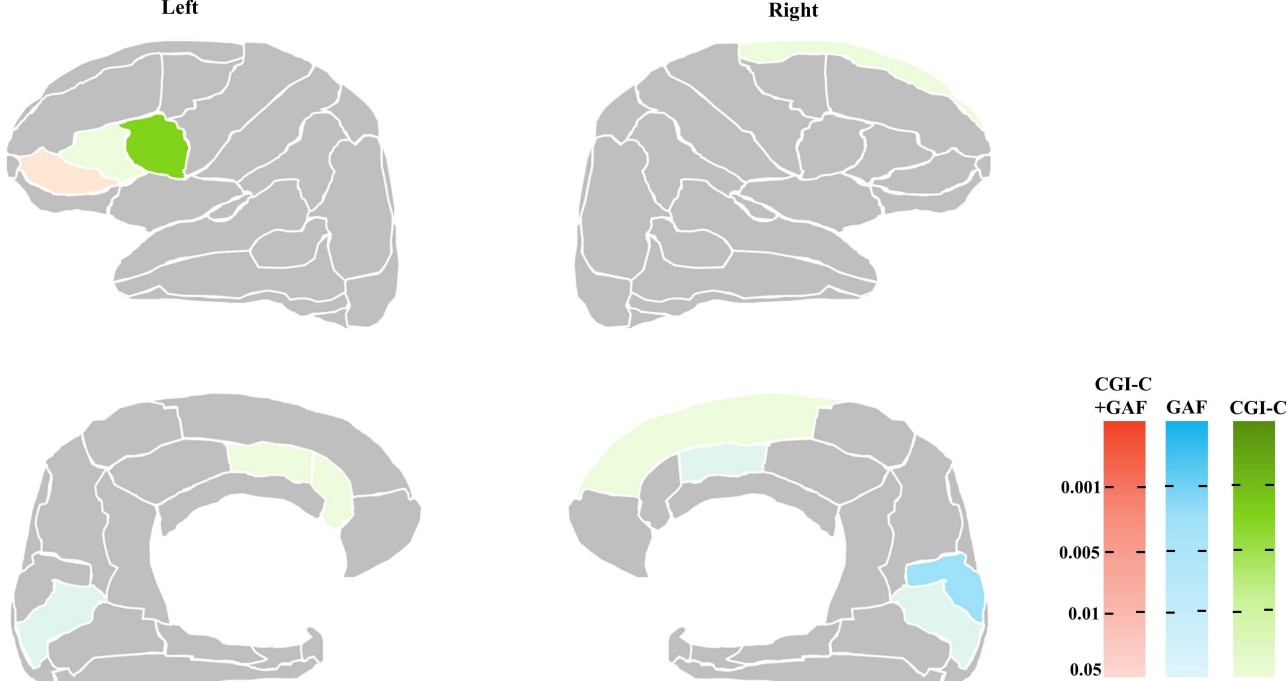

**Fig. 4. The brain areas related to responsiveness to clozapine.** The different colors indicate whether the significant areas are related to ΔGAF (*blue*), CGI-C (*green*) or both (*red*). **Left panels:** the lateral (*upper*) and medial (*lower*) surfaces of the left hemisphere. **Right panels:** the lateral (*upper*) and medial (*lower*) surfaces of the right hemisphere.

**Table 2. Cortical areas that may be relevant to the CLZ response (with uncorrected p<0.05).**

| Region | Clinical parameters | Statistic values |
|---|---|---|
| Lt caudal anterior cingulate | CGI-C | r=0.545, **p=0.029** |
| Rt caudal anterior cingulate | GAF change | r=0.544, **p=0.030** |
| Lt rostral anterior cingulate | CGI-C | r=0.522, **p=0.038** |
| Lt pars orbitalis | GAF change | r=0.563, **p=0.023** |
| | CGI-C | r=0.499, **p=0.049** |
| Lt pars triangularis | CGI-C | r=0.610, **p=0.012** |
| Lt pars opercularis | CGI-C | r=0.665, **p=0.005** |
| Rt superior frontal | CGI-C | r=0.546, **p=0.029** |
| Rt cuneus | GAF change | r=0.688, **p=0.003** |
| Lt pericalcarine | GAF change | r=0.613, **p=0.012** |
| Rt pericalcarine | GAF change | r=0.549, **p=0.028** |

Only the regions with significant differences are shown.

The analysis included age, sex, and MRI scanner as covariates.

patients with TRS were likely to have been influenced by the study's small sample size, requiring a cautious interpretation of our findings.

Several MRI studies of individuals with TRS reported widespread structural alterations in the frontal and temporal lobes as well as the insular, parahippocampal, parietal, and occipital cortices compared to non-TRS patients or healthy

individuals [32–34]. Our present study's ANCOVA of the three groups (controlling for age and sex) demonstrated that compared to the healthy control group, the TRS and non-TRS patients had a significantly smaller cortical volume in 56 of the 68 brain regions examined, ranging from the frontal/temporal to parietal/occipital cortices (**S5 Table** and **Fig 3**), which is consistent with the findings of previous studies. However, we detected no significantly different region between our TRS and non-TRS groups at the rigorous threshold with FDR correction in the three-group or two-group comparisons, although the regional volumes of the lateral/medial frontal, lateral temporal, and occipital cortices differed significantly between these groups at the uncorrected level (**S3**, **S4 Tables** and **Fig 2**). Several research groups reported that relative to non-TRS patients, TRS patients had a smaller regional volume or cortical thickness in the lateral frontal cortices [24,25,35–40], the medial frontal cortices [24], the lateral temporal cortices [24,25,38,40–42], the medial temporal cortices [36], the parietal cortices [38,41], the occipital cortices [24,25,37,40,41], and the insula [36,39].

Our TRS group exhibited middle-to-large effect sizes of smaller cortical volume in multiple areas compared to the HC group in the volume analysis (**S5 Table**). The reason for the almost complete absence of a significant region in the volume ratio analysis (**S3 Table**) is most likely that the whole brain-volume (ICV) was decreased, resulting in the attenuation of an actual decrease in a regional volume in the patient groups. It thus appears that the brain structural alterations in our TRS patients spread widely to broad cortical areas, which is consistent with the findings of earlier investigations.

Regarding the differences we observed between the present CLZ responders and non-responders, several research groups have reported a smaller regional volume or cortical thickness in non-responders to CLZ in the dorsolateral prefrontal cortex (DLPFC) [36], medial frontal cortices [19], and temporal cortices [22,43] compared to CLZ responders, whereas other researchers reported no regional differences [24,25] or no regions related to symptom improvements [44]. These differences in brain structure between CLZ responders and non-responders have been uncertain in part because few such studies have been conducted and each study had a small sample size. Most of the previous studies cross-sectionally compared their patients based on the response to CLZ at a timepoint after CLZ had been administered for a certain duration, which might also influence the findings to some degree since CLZ can affect the brain structure [19, 44, 45]. In the present study, we used MRI findings that had been obtained just prior to the introduction of CLZ treatment. However, our analyses did not detect any clear differences in cortical areas related to the patients' CLZ response, suggesting difficulty in predicting the CLZ response based only on a specific regional structure. In addition, other factors such as relatively severe psychopathology or longer illness duration in our non-TRS group might have contributed to the absence of between-group differences.

Nevertheless, this study detected several cortical regions that might be related to patients' responsiveness to CLZ at the uncorrected level for significance. The left IFG (i.e., pars orbitalis, pars triangularis, and pars opercularis), the bilateral ACC, and the occipital cortex formed clusters relevant to the CLZ response: these regions were larger in the R-CLZ patients than in the NonR-CLZ patients, and a larger volume in these areas was related to a better response to CLZ (**Table 2**).These regions (other than the occipital cortex [pericalcarine]) exhibited effect sizes that were similar in the TRS and non-TRS groups (**S5 Table**), indicating that the structures in these regions are more vulnerable to schizophrenia itself than its severity.

Alteration in the ACC has been observed in several TRS neuroimaging studies: a smaller regional volume or cortical thickness in TRS patients [22,42,46] or CLZ non-responders [22,25], and a higher glutamate or glutamate-glutamine (Glx) level in patients with TRS [47–49] or ultra-treatment-resistant schizophrenia (UTRS) [50]. Other studies have demonstrated that the IFG and occipital cortex in TRS patients showed a regional volume reduction or cortical thinness compared to healthy controls [22,24,35–37,39,42,46] or non-TRS patients [24,25,38,40,41]. Of note, since these findings were significant only at the uncorrected level, they should be interpreted with caution. In addition, these subtle differences in our TRS patients may have contributed to this study's overall negative findings regarding the patients' responses to CLZ. These regions detected in the analyses have been implicated in pathology of TRS, but we were unable to reach a firm conclusion about this with our present results.

Contrary to our initial hypothesis, no cortical area had a significant correlation with the length of time prior to CLZ introduction in this study. In addition, in our analyses, the concept of the clinical importance of the length of time prior to the introduction of CLZ might not have come into existence since we did not observe a significant relationship between this indicator and our patients' responses to CLZ as measured by the change in their GAF scores (i.e., our primary outcome defined in advance), although we observed a significant negative relationship between this measure and the CGI-C scores (i.e., the longer the length of time before CLZ introduction, the less the patients' symptoms improved) (**S1 File**). This result is partly consistent with those of earlier reports and supports the clinical importance of the length of time prior to the introduction of CLZ. The lack of a correlation between this measure and the patients' GAF scores indicates that the length of time prior to CLZ introduction might be an inadequate indicator to search for brain regions relevant to CLZ responsiveness, thus leading to the negative finding of no correlation between this indicator and any cortical region in the patients.

We have found no published investigation of the potential relationship between the length of time before CLZ introduction and cortical regions related to schizophrenia patients' responses to CLZ treatment. It thus remains difficult to identify the reason(s) for our negative findings. One possible reason is that our CLZ patients had relatively longer delays in CLZ commencement (~4 years in the R-CLZ patients and ~5 years in the NonR-CLZ patients) (**S2 Table**), which may have led to damage to the brain to such a degree that a meaningful relationship was lost due to the ceiling effect. A second potential reason is that the regional structural alterations may have varied among the patients, and/or the pace of regional-volume reduction does not harmonize within an individual's brain; these could be influenced by age, disease state and course, medications and more. These deviations in the progressive volume reduction among brain regions or among patients may make it difficult to identify only the cortical areas that are relevant to the limited period of time from the point at which the patients met the TRS criteria to the introduction of CLZ treatment.

However, our findings did not indicate that the length of time prior to CLZ introduction is not a meaningful measure in the treatment of individuals with TRS. The use of non-CLZ agents although the patient meets the criteria for TRS is likely to continue as a use of high-dose antipsychotics, leading to vulnerability to a relapse episode of psychosis [9,11], which could affect brain structural damages [20]. The period from meeting the TRS diagnostic criteria to the commencement of CLZ treatment could affect relevant factors other than brain structures, and these factors may influence each other; combined effects of factors might also affect brain structural alterations and responsiveness to CLZ. Therefore, in efforts to identify brain regions that are relevant to the length of time before the introduction of CLZ, a longitudinal study is warranted to gain a firm conclusion.

There are numerous study limitations to consider. The first is the small number of patients with TRS and patients treated with CLZ in particular. Examinations of a larger number of TRS patients may change the several negative results of this study to positive results. In addition, we analyzed MRI images that were scanned for clinical use but not for research. There might have been many patients who were scanned at other different timepoints or who could not undergo an MRI examination due to their severe psychopathology (prior to the introduction of CLZ in particular). These methodological and clinical circumstances contributed to the small number of patients.

A second study limitation is the insufficient clinical measures since we systematically assessed the psychopathology and functioning of patients with TRS alone; we did not assess these for our non-TRS patients since most of the patients in this group were followed in outpatient settings. Regarding the diagnosis of TRS, the validity of the present patients' TRS diagnoses might have been less accurate since the patients' adherence to medication was confirmed only by the physicians' observations: specifically, this is true of the patients who were classified as having TRS but were not administered CLZ. This inconsistent data collection and somewhat rough diagnosis circumstances hindered rigorous analyses.

A third study limitation is that the dose of CLZ in the patients who were still being treated with CLZ at 1 year was relatively low. It has been suggested that East Asian patients can achieve clinical improvements with a lower CLZ dose compared to patients of European and West Asian ancestry, possibly based on ethnic differences in the ability to metabolize

CLZ [51]. Our patients' CLZ doses were within the ranges reported for Japanese patients. However, in our group of NonR-CLZ patients, higher CLZ doses might have enhanced the response to CLZ to some degree. Their clinical scores might also have been influenced by the assessments having been conducted by different physicians as well.

A fourth study limitation is that the MRI images were obtained with five different MRI scanners. For the group of healthy controls, we turned to the open dataset of healthy participants based on MRI findings and on protocols that differed from those used for our patients. This somewhat complicated situation led to difficulty in achieving standard harmonization such as that provided by the ComBat method, thus providing a serious study limitation. We thus used the relative ratio of volume (regional volume per ICV) and the MRI scanners as covariates for the comparisons with the healthy controls' data and several analytic procedures. However, since such counterplans could not overcome all of the technical drawbacks, our overall results are preliminary and warrant replication studies.

## Conclusions

In this study of a fully clinical-based population, the patients with TRS exhibited widespread cortical volume reductions (mainly in the frontal, temporal, parietal, and occipital cortices) relative to the healthy controls, whereas only subtle differences were observed between the TRS and non-TRS patients. Differences in cortical structure between CLZ responders and non-responders were not observed at a stringent statistical threshold; however, several regions (IFG, ACC, and occipital cortex) were identified in between-group and correlational analyses at an uncorrected level. The length of time prior to the introduction of treatment with CLZ did not show a significant relationship with any brain region. These findings were affected by the small sample size but suggest that areas implicated in the pathology of schizophrenia could be involved in the response to CLZ. Further studies with larger numbers of patients and a prospective research design are warranted.

## Supporting information

**S1 File. Supplementary materials.**
(DOCX)

**S1 Fig. Scatter plots showing the relationship between brain volume and age by different MRI scanners for (A) patients and (B) healthy controls.**
(TIF)

**S2 Fig. Scatter plots showing volume distributions in (A) the left rACC and (B) eTIV, and (C) rACC volume ratio (derived by A/B) by different MRI scanners in the TRS, non-TRS and HC groups.** The horizontal line represents the mean of each group.
(TIF)

**S3 Fig. The areas of greater cortical volume reduction in clozapine non-responsers compared to clozapine responsers (at the uncorrected p<0.05).**
(TIF)

**S1 Table. The parameters of the MRI systems used for the present study.**
(DOCX)

**S2 Table. The TRS patients' demographic and treatment information.**
(DOCX)

**S3 Table. The comparison of the cortical volume and the volume ratio between the TRS and nonTRS groups.**
(DOCX)

**S4 Table. The comparison of cortical volume between the TRS and non-TRS groups in the analysis including the patients' age at illness onset and the antipsychotic dose as additional covariates.**
(DOCX)

**S5 Table. Comparison of the cortical volume ratio in the TRS, non-TRS, and HC groups.**
(DOCX)

**S6 Table. Comparison of cortical volume ratios between the CLZ responders and CLZ non-responders.**
(DOCX)

**S7 Table. Cortical areas potentially relevant to the CLZ response in the analysis including the antipsychotic dose as an additional covariate.**
(DOCX)

## Author contributions

**Conceptualization:** Yuto Masumo, Nobuhisa Kanahara, Hiroshi Komatsu, Masaomi Iyo.

**Data curation:** Yuto Masumo, Nobuhisa Kanahara, Yuji Otsuka, Teruyuki Ishii, Hirotaka Sato, Keita Idemoto.

**Formal analysis:** Yuto Masumo, Nobuhisa Kanahara, Yoshiyuki Hirano.

**Funding acquisition:** Nobuhisa Kanahara.

**Investigation:** Yuto Masumo, Nobuhisa Kanahara.

**Methodology:** Yuto Masumo, Nobuhisa Kanahara, Yusuke Sudo, Yoshiyuki Hirano.

**Project administration:** Nobuhisa Kanahara.

**Resources:** Yuto Masumo, Nobuhisa Kanahara, Yuji Otsuka, Keita Idemoto, Yasunori Oda, Tomihisa Niitsu.

**Software:** Yuto Masumo, Nobuhisa Kanahara, Yusuke Sudo, Yoshiyuki Hirano.

**Supervision:** Hiroshi Komatsu, Yuko Fujita, Yoshiyuki Hirano, Masaomi Iyo.

**Validation:** Nobuhisa Kanahara, Yusuke Sudo, Yoshiyuki Hirano.

**Visualization:** Yuto Masumo, Yusuke Sudo.

**Writing – original draft:** Yuto Masumo, Nobuhisa Kanahara.

**Writing – review & editing:** Nobuhisa Kanahara, Yoshiyuki Hirano.

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
