## [Decision Letter · Decision Letter 0]

1 Dec 2025

Dear Dr. Kanahara,

Thank you for submitting your manuscript to PLOS ONE. After careful consideration, we feel that it has merit but does not fully meet PLOS ONE’s publication criteria as it currently stands. Therefore, we invite you to submit a revised version of the manuscript that addresses the points raised during the review process.

We look forward to receiving your revised manuscript.

Kind regards,

Kenji Tanigaki, Ph.D., M.D.

Academic Editor

PLOS ONE

Journal Requirements:

3. Thank you for stating the following in the Competing Interests/Financial Disclosure * (delete as necessary) section:

“N.K. reports honoraria from Otsuka Pharmaceutical Co., Sumitomo Pharma Co., Janssen Pharmaceutical., Meiji Seika Pharma Co., Mitsubishi Tanabe Pharma Co., MSD. Y.Od. reports honoraria from Eisai Co., Otsuka Pharmaceutical Co., Meiji Seika Pharma Co. T.N. reports honoraria from Otsuka Pharmaceutical Co., Viatris Inc., Daiichi Sankyo Co., Sumitomo Pharma Co., Meiji Seika Pharma Co., and Yoshitomi Yakuhin Co. He also received research funding from NTT Precision Medicine, Inc. Y.M., Y. Ot., Y.S., T.I., H.S., K.I., H.K., Y.F., Y.H., and M.I. have no conflicts of interest to declare.”

We note that you received funding from a commercial source: “Otsuka Pharmaceutical Co., Sumitomo Pharma Co., Janssen Pharmaceutical., Meiji Seika Pharma Co., Mitsubishi Tanabe Pharma Co. Otsuka Pharmaceutical Co., Viatris Inc., Daiichi Sankyo Co., Sumitomo Pharma Co., Meiji Seika Pharma Co., and Yoshitomi Yakuhin Co”

4. We note that your Data Availability Statement is currently as follows:” All relevant data are within the manuscript and its Supporting Information files.”

If your submission does not contain these data, please either upload them as Supporting Information files or deposit them to a stable, public repository and provide us with the relevant URLs, DOIs, or accession numbers. For a list of recommended repositories, please see https://journals.plos.org/plosone/s/recommended-repositories .

Reviewers' comments:

Reviewer's Responses to Questions

**Comments to the Author**

1. Is the manuscript technically sound, and do the data support the conclusions?

Reviewer #1: No

Reviewer #2: Partly

2. Has the statistical analysis been performed appropriately and rigorously?

Reviewer #1: No

Reviewer #2: Yes

3. Have the authors made all data underlying the findings in their manuscript fully available?

Reviewer #1: Yes

Reviewer #2: No

4. Is the manuscript presented in an intelligible fashion and written in standard English?

Reviewer #1: No

Reviewer #2: No

Reviewer #1: The manuscript investigates cortical volumetric alterations in patients with treatment-resistant schizophrenia (TRS), and explores whether structural MRI findings are associated with response to clozapine (CLZ). The study leverages clinically acquired MRI data (n = 104 patients, plus 109 external healthy controls) and FreeSurfer-based cortical parcellation according to the Desikan–Killiany atlas.

Altough the topic is clinically relevant, the study suffers from significant methodological and interpretative limitations that limit the robustness and novelty of its conclusions. The results are largely negative or nominal, and the discussion sometimes overstates their significance.

Overall, in my opinion the manuscript does not meet the miniml publication standards and should be rejected, moved to a lower impact factor journal, or modified as a brief report with preliminary exploratory nature. I attach below a list of my major concerns.

1. The total number of TRS patients treated with CLZ is small (n=20, with only 12 responders and 8 non-responders). This precludes adequate power for voxel- or region-based analyses, especially when correcting for multiple comparisons. Indeed, none of the reported associations survived FDR correction. Therefore, the interpretation of uncorrected results shoud be presented with strong caution or omitted. Moreover, I advise the authors to explicitly acknowledge the exploratory nature of the analyses and avoid causal or mechanistic interpretations

2. MRI scans were obtained from five different scanners and protocols, and the healthy control (HC) group was drawn from an external public dataset acquired on different systems. Although the authors included scanner type as a covariate and used ratio normalization, this ususally does not completely correct for inter-scanner variability. Authors should try to run harmonization methods or discuss this issue as a serious limitation that prevents both generalizability and reliable interpretation of data.

3. No correction for antipsychotic dose or age of onset (and therefore duration of illness) is given, altough these parameters are significantly different between groups and could represent strong confounders for structural brain abnormalities. This precludes any reliable inference on outcomes. And the same attains for the sub-classification in clozapine-responders and non-responders, although the reliability of this analysis is already prevented from the very low number of cases.

4. The operationalization of TRS and the criteria for defining clozapine response/non-response should be described in more detail in the main text (not only in Supplementary Material). The use of clinical global impression (CGI-C) and GAF change over 1 year is reasonable, although not the standard, but the reliability of retrospective classification should be discussed. Moreover, I would avoid to make comparisons between R-CLZ and nR-CLZ patients, given the extremely low sample size in this subclassification and its propensity to false outcomes with this limited power. In my opinion, the section on the comparison between clozapine responders and non-responders should be eliminated as it has poor or no scientific value.

5. The central hypothesis that cortical morphology correlates with the delay before CLZ introduction was not confirmed. This negative finding, which contradicts the stated rationale, should be given greater prominence in the abstract and discussion, rather than being overshadowed by nominal findings on other regions. Moreover, it is not clear on which literature base the authors posited this hypothesis. And in general, the introduction is not useful to capture the rationale of the study or a clear neurobiologically-based hypothesis, other than assuming that there are few previous reports on the issue. And the strong claim on the evaluation of biomarkers related to clozapine therapy should be mitigated, given than only 20 patients of the sample were under clozapine

6. The discussion is in large part speculative, as it repeatedly refers to “involvement” of IFG, ACC, and occipital cortex in CLZ response, yet all findings are at p < 0.05 uncorrected. These findings should be clearly labelled as exploratory and interpreted accordingly, with great caution and no causal inferences

7. The choice to analyze cortical volume ratios rather than absolute values or thickness measures is unconventional and not well justified. The authors should clarify why this approach was preferred and discuss its possible drawbacks.

8. The manuscript would benefit from linguistic editing to improve clarity and conciseness. Several sections are redundant or overly descriptive. Figures and tables are informative but should highlight key results more effectively (e.g., provide effect sizes and FDR-adjusted p-values in the main text).

9. Authors should clarify the time relationship between MRI acquisition and the initiation of clozapine treatment (was the MRI always before CLZ introduction or any other else?). The ethics section should specify the form of consent (written vs. opt-out) for both institutions.

Reviewer #2: This study investigates grey matter volumen correlates to TRS and CPZ repsonse in patients with schizophrenia. The study questions are interesting and clinically relevant, and the authors aim to answer the research questions based on pragmatic solutions on data-quality, sample-size, methodological complexity a.o.. There are however a range of major and minor concerns about the study:

1. It is a major concern that the study and data acquisition was not designet for these hypothesis, but was derived from a pragmatic pooling of data from 7 different scanners with vital differences in acquisition parameters such as Field Strenght, REp. time, Inversion, flip angle etc. (according to Supplem. Table S1). The authors state as imitations that harmonization attempts failed, hence they used the very crude correction for scanner type as covariate. As the major results reported is groupdiffernce between patients and controls, where the MRI data for controls were from a very different public dataset, the results are unfortunately questionable. What could validate the results more would be transparence about an extensive examination of data, including an option of visual inspection of e.g. scatterplots etc. by reader and reviewers. However, as there are more major considerations, such as a lack of power calculation, resulting in a far to small sample (8 non-responsers vs 12 responding CPZ patients when examining groupdifferences) to meaningfully answer the relevant research questions. The authors may - with the caveats from the pooling of data from different scanners - meaningfully investigate differences between controls and patients, but further subgrouping of patients will reduce the degrees of freedom to a point where analyses cannot be evaluated as sound / nor results as robust with this sample size. It is a huge concern with yet another MRI study presenting non-robust results on insufficient data of limited quality. The authors state (line 228-229) that they provide several insights into the relationship between brain structure and responsiveness to CPZ - but this conclusion is not valid. At best, the study may be hypothesis generating due to the pragmatic design and major concerns.

2. Another major concern is the attempt to link MRI GMV data to CPZ response. First, it is very uncertain that any difference has an MRI-signal, as the patients were not antipsychotic naive before start CPZ. Secondly, it is not proper to present this mixture of uncorrected (i.e. "nominal level of significance" as in line 182)and corrected results, particularly with the huge number of correlation analyses - a method known for producing to many false positive results. If the authors - after correction for multiple analyses - find that no GM regions survived control when comparing CPZ responders to nonresponders, that is the result which has to be clearly stated in eg. the abstract. It is not, to the contrary it appears like there is a difference, which is not trustworthy.

3. The use of unusual words and terms, which are a bit uncorrect, and claims which demonstrate lack of clinical understanding (such as line 36 "broader brain regions"; line 153 "MRI-system" (usually we term it scanner / scannersites); line 165-167 are not understandable . do the authors correct for different scanners or not in the analyses?; line 301 "cortex was less damaged"; line 286-287, claiming that hallucinations, negative symptoms and cognitive dysfunction is a "potential core psychopathology of TRS" (these are decisive symptoms for all patients with schizophrenia??).

4. Lack of clarity in reporting the study design, materials and methods is a huge obstacle for transparency in this study. The methods reported in supplements needs to go to the main text, and it would be beneficial with a illustration of the analyses plan, and which data came from which group / subgroup / timepoint, as it is hard to deciphre in the current format. Further, it is unclear in Table 2 that you actually appear to have tested every clinical measure at against coorticsl regions, but in some it is delta (change), but not in others etc.

5. Minor concerns would be the lenght of the discussion, with a lot of redundance, such as line 22-225 repeated in 237-239, and it appears unstructured, with far too much speculation on results which cannot justify such discussions.

A major revision, using the principle of Occams razor, where the authors refrain from underpowered analyses, not overreport results, and instead simplify the study questions and enhance clarity and transparency of the pragmatic and preliminary hypothesis-generating results would result in an article of much higher scientific quality.

**Do you want your identity to be public for this peer review?** For information about this choice, including consent withdrawal, please see our Privacy Policy

Reviewer #1: **Yes:** Felice Iasevoli

Reviewer #2: No

---

## [Author Response · Author response to Decision Letter 1]

10 Jan 2026

Revised ms. PONE-D-25-50031: "Neural substrates of treatment-resistant schizophrenia and the response to clozapine A structural MRI study in a clinical setting"

Response to the Journal Editor

Journal Requirements:

Comment 1. Please ensure that your manuscript meets PLOS ONE's style requirements, including those for file naming. The PLOS ONE style templates can be found at

Response: We have ensured that the manuscript meets the above additional style requirements.

Comment 2. Please provide additional details regarding participant consent. In the ethics statement in the Methods and online submission information, please ensure that you have specified (1) whether consent was informed and (2) what type you obtained (for instance, written or verbal, and if verbal, how it was documented and witnessed). If your study included minors, state whether you obtained consent from parents or guardians. If the need for consent was waived by the ethics committee, please include this information.

Response: Our study had a retrospective design and we analyzed MRI images that had been scanned for clinical use in the past, but not for research purposes. Regarding study ethics, we note in the manuscript that the study followed the Ethical Guidelines for Medical and Biological Research Involving Human Subjects in Japan. Regarding informed consent, we advertised the study on the two hospitals' websites, providing an opportunity to opt out (described in the manuscript text). We created an independent 'Ethics review' subsection in the Patients and Methods section on page 8.

Comment 3. Thank you for stating the following in the Competing Interests/Financial Disclosure * (delete as necessary) section:

“N.K. reports honoraria from Otsuka Pharmaceutical Co., Sumitomo Pharma Co., Janssen Pharmaceutical., Meiji Seika Pharma Co., Mitsubishi Tanabe Pharma Co., MSD. Y.Od. reports honoraria from Eisai Co., Otsuka Pharmaceutical Co., Meiji Seika Pharma Co. T.N. reports honoraria from Otsuka Pharmaceutical Co., Viatris Inc., Daiichi Sankyo Co., Sumitomo Pharma Co., Meiji Seika Pharma Co., and Yoshitomi Yakuhin Co. He also received research funding from NTT Precision Medicine, Inc. Y.M., Y. Ot., Y.S., T.I., H.S., K.I., H.K., Y.F., Y.H., and M.I. have no conflicts of interest to declare.”

We note that you received funding from a commercial source: “Otsuka Pharmaceutical Co., Sumitomo Pharma Co., Janssen Pharmaceutical., Meiji Seika Pharma Co., Mitsubishi Tanabe Pharma Co. Otsuka Pharmaceutical Co., Viatris Inc., Daiichi Sankyo Co., Sumitomo Pharma Co., Meiji Seika Pharma Co., and Yoshitomi Yakuhin Co”

Response: This study was not supported by any pharmaceutical company. All of the Competing Interest contents are honoraria that are irrelevant to the study but are shown to provide transparency as follows:

“NK reports honoraria from Otsuka Pharmaceutical Co., Sumitomo Pharma Co., Janssen Pharmaceutical., Meiji Seika Pharma Co., Mitsubishi Tanabe Pharma Co., MSD. YOd reports honoraria from Eisai Co., Otsuka Pharmaceutical Co., Meiji Seika Pharma Co. TN reports honoraria from Otsuka Pharmaceutical Co., Viatris Inc., Daiichi Sankyo Co., Sumitomo Pharma Co., Meiji Seika Pharma Co., and Yoshitomi Yakuhin Co. He also received research funding from NTT Precision Medicine, Inc. YM, YOt, YS, TI, HS, KI, HK, YF, YH, and MI have no conflicts of interest to declare. This does not alter our adherence to PLOS ONE policies on sharing data and materials.”

Comment 4. We note that your Data Availability Statement is currently as follows:” All relevant data are within the manuscript and its Supporting Information files.”

Response: We think that our manuscript already met these requests.

Comment 5. If the reviewer comments include a recommendation to cite specific previously published works, please review and evaluate these publications to determine whether they are relevant and should be cited. There is no requirement to cite these works unless the editor has indicated otherwise.

Response: The two Reviewers' comments did not include a request to add another published article.

Revised ms. PONE-D-25-50031: "Neural substrates of treatment-resistant schizophrenia and the response to clozapine A structural MRI study in a clinical setting"

Response to Reviewer#1

Reviewer #1:

The manuscript investigates cortical volumetric alterations in patients with treatment-resistant schizophrenia (TRS), and explores whether structural MRI findings are associated with response to clozapine (CLZ). The study leverages clinically acquired MRI data (n = 104 patients, plus 109 external healthy controls) and FreeSurfer-based cortical parcellation according to the Desikan–Killiany atlas.

Altough the topic is clinically relevant, the study suffers from significant methodological and interpretative limitations that limit the robustness and novelty of its conclusions. The results are largely negative or nominal, and the discussion sometimes overstates their significance.

Overall, in my opinion the manuscript does not meet the miniml publication standards and should be rejected, moved to a lower impact factor journal, or modified as a brief report with preliminary exploratory nature. I attach below a list of my major concerns.

Response: We appreciate the many valuable comments on our work, and we completely agree with all of the comments and accept the Reviewer's concerns. We have made major corrections. New text is presented in red font, and the text exchanged from the Supplementary Material to the main text and vice versa is presented in blue font.

Comment 1. The total number of TRS patients treated with CLZ is small (n=20, with only 12 responders and 8 non-responders). This precludes adequate power for voxel- or region-based analyses, especially when correcting for multiple comparisons. Indeed, none of the reported associations survived FDR correction. Therefore, the interpretation of uncorrected results shoud be presented with strong caution or omitted. Moreover, I advise the authors to explicitly acknowledge the exploratory nature of the analyses and avoid causal or mechanistic interpretations

Response: We revised the text to provide a strict statistically significant threshold more clearly and consistently throughout the manuscript. To this end, the following two revisions were made for the comparison of the R-CLZ (n=12) and NonR-CLZ (n=8) groups.

1. Although this comparison was initially planned as the study's main analysis, we were unable to collect a sufficient number of patient cases for this comparison. In the Results section, we now state that we collected a smaller sample than initially expected, and we moved the description of this from the main text to the Supplementary Material.

2. Throughout the revised manuscript, the results of the analyses on clozapine response (both the between-group comparison and the correlational analysis) with the uncorrected p<0.05 (nominal-level of significance) are described with the negative findings.

These revisions are in the Abstract, Results and Discussion as follows.

Page 2, Lines 38-40 in the Abstract

“The correlational analysis of regions related to clozapine responsiveness identified the inferior frontal gyrus (IFG), anterior cingulate cortex (ACC), and occipital cortex at the uncorrected level of significance, but none of these survived the correction for multiple comparisons.”

Page 21, Lines 281-285 in the Results

“Since this comparison included a smaller number of patients than initially planned (R-CLZ, n=12; NinR-CLZ, n=8), the results are provided in the Supplementary Material and S6 Table. Briefly, no significantly different cortical volume region was observed at the statistical threshold after FDR correction in the multiple comparison.”

Page 24, Lines 321-325 in the Discussion

“Regarding the responsiveness to CLZ, no regions survived the FDR correction for multiple-group comparisons: we observed that several regions including the inferior frontal gyrus (IFG), ACC, and occipital cortex were related to the CLZ response in the between-group comparison or in the correlational analysis at the uncorrected level for significance (Table 2, S6, S7 Tables and Fig 4, S1 Fig).”

Page 28, Lines 380-385 in the Discussion

“Of note, since these findings were significance only at the uncorrected level, they should be interpreted with caution. In addition, these subtle differences in our TRS patients may have contributed to this study's overall negative findings regarding the patients' response to CLZ. These regions have been implicated in the pathology of TRS, but we were unable to reach a firm conclusion about this with our present results.”

Comment 2. MRI scans were obtained from five different scanners and protocols, and the healthy control (HC) group was drawn from an external public dataset acquired on different systems. Although the authors included scanner type as a covariate and used ratio normalization, this ususally does not completely correct for inter-scanner variability. Authors should try to run harmonization methods or discuss this issue as a serious limitation that prevents both generalizability and reliable interpretation of data.

Response: We agree that the analyses without harmonization were a major study limitation, and we have now described this point in the Discussion as follows.

Page 32, Lines 445-447 in the Discussion

“This somewhat complicated situation led to difficulty in achieving standard harmonization such as that provided by the ComBat method, thus providing a serious study limitation.”

Comment 3. No correction for antipsychotic dose or age of onset (and therefore duration of illness) is given, altough these parameters are significantly different between groups and could represent strong confounders for structural brain abnormalities. This precludes any reliable inference on outcomes. And the same attains for the sub-classification in clozapine-responders and non-responders, although the reliability of this analysis is already prevented from the very low number of cases.

Response: We have added the results controlling for the antipsychotic dose and the patient age at onset in addition to the original variables (i.e., age, sex, MRI) for the comparison of the TRS and non-TRS groups. For the analysis for clozapine response (both the between-group comparison and correlational analysis), we calculated the data including the antipsychotic dose as an additional covariate. These analyses overall obtained almost the same results, which are now presented as novel S4, S6 and S7 Tables.

Page 17, Lines 232-234 in the Results

“When the patients' age at illness onset and the antipsychotic dose were included as additional covariates, the analysis revealed almost the same results (S4 Table).”

Page 21, Lines 281-283 in the Results

“Since this comparison included a smaller number of patients than initially planned (R-CLZ, n=12; NonR-CLZ, n=8), the results are provided in the Supplementary Material (2. Comparison of cortical volume between the CLZ responders and non-responders) and S6 Table”

Page 21, Lines 295-296 in the Results

“These results were almost the same when the analysis included the antipsychotic dose as a covariate (S7 Table).”

Comment 4. The operationalization of TRS and the criteria for defining clozapine response/non-response should be described in more detail in the main text (not only in Supplementary Material). The use of clinical global impression (CGI-C) and GAF change over 1 year is reasonable, although not the standard, but the reliability of retrospective classification should be discussed. Moreover, I would avoid to make comparisons between R-CLZ and nR-CLZ patients, given the extremely low sample size in this subclassification and its propensity to false outcomes with this limited power. In my opinion, the section on the comparison between clozapine responders and non-responders should be eliminated as it has poor or no scientific value.

Response: We appreciate these comments. As you pointed out, there could be less accuracy concerning the diagnosis of TRS for patients who did not take clozapine since their drug-adherence level were evaluated by their physicians in clinical practice. In contrast, for the TRS patients who were being treated with clozapine, their diagnosis is more valid since physicians are required to confirm the adherence level of their patients prior to the introduction of clozapine. However, even in patients taking clozapine, there might be some problems with the assessments of the clozapine response varying among different physicians. These limitations were derived from our study's retrospective design and were added to the study limitations section. In accord with your instruction, we moved the detailed descriptions of the methodical procedure including the diagnosis and assessments in the Supplementary Material back to the main text (Methods section).

Regarding the results of the between-group comparis

---

## [Decision Letter · Decision Letter 1]

2 Feb 2026

Dear Dr. Kanahara,

We look forward to receiving your revised manuscript.

Kind regards,

Kenji Tanigaki, Ph.D., M.D.

Academic Editor

PLOS One

Journal Requirements:

Reviewers' comments:

Reviewer's Responses to Questions

**Comments to the Author**

Reviewer #1: All comments have been addressed

Reviewer #2: (No Response)

2. Is the manuscript technically sound, and do the data support the conclusions?

Reviewer #1: Yes

Reviewer #2: Partly

3. Has the statistical analysis been performed appropriately and rigorously?

Reviewer #1: Yes

Reviewer #2: No

4. Have the authors made all data underlying the findings in their manuscript fully available?

Reviewer #1: Yes

Reviewer #2: (No Response)

5. Is the manuscript presented in an intelligible fashion and written in standard English?

Reviewer #1: Yes

Reviewer #2: Yes

Reviewer #1: The authors have addressed most of my concerns and appropriately toned down their conclusions. While important methodological limitations remain, the manuscript now presents a clearer and more transparent exploratory analysis that is suitable for publication in PLOS ONE.

Reviewer #2: The authors have not adressed my concerns sufficiently. The first major concern reg. data acquisition from a pragmatic pooling of data from 7 different scanners with vital differences in acquisition parameters where authors state that harmonization attempts failed, plus the MRI data for controls were from a very different public dataset. I suggested

" more transparence about an extensive examination of data, including an option of visual inspection of e.g. scatterplots etc. by reader and reviewers", but the only action taken by the authors is to move text from supplementary to main text. I would like a thorough examination of how attempts to harmonize was failing, including the results as shown by eg.scatterplots etc. - if not, the conclusions cannot be regarded as valid.

My second concern "a lack of power calculation, resulting in a far to small sample (8 non-responsers vs 12 responding CPZ

patients when examining groupdifferences) to meaningfully answer the relevant research questions. The authors may - with the caveats from the pooling of data from different scanners - meaningfully investigate differences between controls and patients, but further subgrouping of patients will reduce the degrees of freedom to a point where analyses cannot be evaluated as sound / nor results as robust with this sample size." was solved by authors by moving the subgroup analyses to the supplementary and change some text - but still in abstract authors write predominantly about this subject - in results: "The correlational analysis of regions related to clozapine responsiveness identified the inferior frontal gyrus (IFG), anterior cingulate cortex (ACC), and the occipital cortex at the uncorrected level of significance, but none of these survived t he correction for multiple comparisons" and in Conclusion "These results suggest that the IFG, ACC, and occipital cortex might be responsible for the treatment response to clozapine....". It is very difficult to infer from what could have been a result in case of more power, but here I truly find this to be scientifically unsound, and not just something that be framed differently by small reformulaitons. I cannot recommend accept without thorough rewriting from a much more humble perspective on what is possible to infer by this study design and underpowered samplesize.

**Do you want your identity to be public for this peer review?** For information about this choice, including consent withdrawal, please see our Privacy Policy

Reviewer #1: No

Reviewer #2: No

---

## [Author Response · Author response to Decision Letter 2]

16 Feb 2026

Revised ms. PONE-D-25-50031R2: "Neural substrates of treatment-resistant schizophrenia and the response to clozapine A structural MRI study in a clinical setting"

Responses to Reviewer 1

Reviewer 1:

The authors have addressed most of my concerns and appropriately toned down their conclusions. While important methodological limitations remain, the manuscript now presents a clearer and more transparent exploratory analysis that is suitable for publication in PLOS ONE.

Response: Thank you. We appreciate your helpful input and review of our manuscript.

Revised ms. PONE-D-25-50031R2: "Neural substrates of treatment-resistant schizophrenia and the response to clozapine A structural MRI study in a clinical setting"

Responses to Reviewer 2

Reviewer 2:

Comment 1. The authors have not adressed my concerns sufficiently. The first major concern reg. data acquisition from a pragmatic pooling of data from 7 different scanners with vital differences in acquisition parameters where authors state that harmonization attempts failed, plus the MRI data for controls were from a very different public dataset. I suggested

" more transparence about an extensive examination of data, including an option of visual inspection of e.g. scatterplots etc. by reader and reviewers", but the only action taken by the authors is to move text from supplementary to main text. I would like a thorough examination of how attempts to harmonize was failing, including the results as shown by eg.scatterplots etc. - if not, the conclusions cannot be regarded as valid.

Response: We are very sorry that we did not adequately address the reviewer’s concerns in our previous revision of our manuscript. In this revised version, we present additional scatter plot data showing individual volume distributions. In order to enhance the data transparency and to provide a resource that could be clearly interpreted for the potential effects of different MRI scanners on brain volume, the overall trend showing the distribution in brain volume (new S1 Figure) and the analysis of the effect of each MRI scanner on regional volume in each group (new S2 Figure) are now presented. A detailed explanation of this analysis is given in the Supplementary Materials as shown below. As a result of this analysis, as the reviewer pointed out, we conclude that specific scanner(s) could potentially affect regional volume for subjects, but our attempt to apply volume-ratio analyses could have attenuated the effect of MRI scanners on between-group differences.

Page 20, Lines 266-269 in the Results

“A further analysis of the potential effects of different MRI scanners on the between-group comparisons was conducted; the results are shown in the Supplementary Materials (2. Potential effects of MRI scanners within each group).”

The 2nd section of the Supplementary Material

“2. Potential effects of MRI scanners within each group

Since the present study did not apply harmonization (ComBat) to correct for potential scanner effects, we examined whether difference among MRI scanners could have influenced the findings.

First, in the patient group, whole-brain volume (“BrainSegVol”) appeared unlikely to be influenced by the five MRI scanners (S1A Fig). Similarly, in the HC group, there were no clear differences in the brain-volume distributions between the two scanners (S1B Fig). However, because the patient and HC groups differed in the MRI scanners used, we further examined whether scanner effects could influence the between-group comparison of brain volume. As an example, we show results for the left rACC (S2 Fig), which exhibited the largest difference among the three groups in the volume-ratio analyses (F=32.595, p=1.0×10-10: S5 Table). In the volume-level comparison, HC volume appeared slightly larger than TRS volume, consistent with the statistical significance by three-group ANOVA (F=8.580, p<0.001; TRS<HC, p<0.001, Bonferroni corrected post-hoc test) (S2A Fig). For eTIV, on the other hand, it is likely that the HC had slightly smaller volume than the non-TRS, which was supported by the three-group ANOVA (F=10.007, p<0.001; nonTRS>HC, p<0.001, Bonferroni correct post-hoc test) (S2B Fig). Notably, within the HC group, participants scanned on MRI-7 showed a trend toward smaller volumes. Together, these observations suggest that volume-level comparisons (particularly comparison vs. HCs drawn from an open dataset) could be influenced by scanner differences, because a global trend toward larger volumes in schizophrenia relative to HCs would be unexpected. When we instead analyzed volume ratios, the separation between HC and patient groups became clearer (S2C Fig; also see the Results section), and the MRI-7-specific trend was no longer apparent. Thus, although scanner-related differences seemed minimal within the patient and HC groups considered separately (S1 Fig), scanner effects may still have influenced between-group comparisons. The volume-ratio approach may help attenuate such effects and may better reflect true between-group differences.”

In relation to these changes, we added two supplemental figures to Supporting information

S1 Fig. Scatter plots showing the relationship between brain volume and age by different MRI scanners for (A) patients and (B) healthy controls.

S2 Fig. Scatter plots showing volume distributions in (A) the left rACC and (B) eTIV, and (C) rACC volume ratio (derived by A/B) by different MRI scanners in the TRS, non-TRS and HC groups. The horizontal line represents the mean of each group.

Comment 2. My second concern "a lack of power calculation, resulting in a far to small sample (8 non-responsers vs 12 responding CPZ patients when examining groupdifferences) to meaningfully answer the relevant research questions. The authors may - with the caveats from the pooling of data from different scanners - meaningfully investigate differences between controls and patients, but further subgrouping of patients will reduce the degrees of freedom to a point where analyses cannot be evaluated as sound / nor results as robust with this sample size." was solved by authors by moving the subgroup analyses to the supplementary and change some text - but still in abstract authors write predominantly about this subject - in results: "The correlational analysis of regions related to clozapine responsiveness identified the inferior frontal gyrus (IFG), anterior cingulate cortex (ACC), and the occipital cortex at the uncorrected level of significance, but none of these survived t he correction for multiple comparisons" and in Conclusion "These results suggest that the IFG, ACC, and occipital cortex might be responsible for the treatment response to clozapine....". It is very difficult to infer from what could have been a result in case of more power, but here I truly find this to be scientifically unsound, and not just something that be framed differently by small reformulaitons. I cannot recommend accept without thorough rewriting from a much more humble perspective on what is possible to infer by this study design and underpowered samplesize.

Response: We agree with the reviewer. In the revised manuscript, we further toned down our interpretation of this study, and we now describe our results as completely negative. We changed the following portions in the Abstract and Conclusion section accordingly.

Page 3, Line 38-Page, 4, Line 39 in the Abstract

“The correlational analysis of regions related to clozapine responsiveness did not identify any region that survived the correction for multiple comparisons.”

Page 4, Lines 41-42 in the Abstract

“Overall, these results failed to identify the cortical region responsible for the treatment response to clozapine.”

Page 33, Lines 457-459 in the Conclusion

“Differences in cortical structure between CLZ responders and non-responders were not observed at a stringent statistical threshold; however, several regions (IFG, ACC, and occipital cortex) were identified in between-group and correlational analyses at an uncorrected level.”

---

## [Editor Report · Decision Letter 2]

3 Mar 2026

Neural substrates of treatment-resistant schizophrenia and the response to clozapine A structural MRI study in a clinical setting

PONE-D-25-50031R2

Dear Dr. Kanahara,

We’re pleased to inform you that your manuscript has been judged scientifically suitable for publication and will be formally accepted for publication once it meets all outstanding technical requirements.

Kind regards,

Kenji Tanigaki, Ph.D., M.D.

Academic Editor

PLOS One

---

## [Editor Report · Acceptance letter]

PONE-D-25-50031R2

PLOS One

Dear Dr. Kanahara,

I'm pleased to inform you that your manuscript has been deemed suitable for publication in PLOS One. Congratulations! Your manuscript is now being handed over to our production team.

Kind regards,

on behalf of

Dr. Kenji Tanigaki

Academic Editor

PLOS One